# BLOCKDECODER: Boosting ASR Decoders with Context and Merger Modules

**Darshan Prabhu**
Department of CSE
IIT Bombay
darshanp@cse.iitb.ac.in

**Preethi Jyothi**
Department of CSE
IIT Bombay
pjyothi@cse.iitb.ac.in

## Abstract

Attention-based encoder decoder models remain a popular choice for state-of-the-art automatic speech recognition (ASR). These models combine a powerful audio encoder that extracts rich acoustic features with a decoder that autoregressively produces the ASR output. The decoder handles two critical tasks: (1) building rich text-only context and (2) merging acoustic information from the encoder to ensure the predictions remain faithful to the audio. We observe a systematic pattern across the attention distributions of decoder layers in prior architectures: the initial layers direct most attention towards building textual context, while the later layers largely focus on merging acoustic and textual information for the final predictions. Leveraging this key insight, we propose BLOCKDECODER, a novel decoder architecture comprising two distinct components: a text encoder that is purely text-based, and a MERGER that combines information from the audio encoder and text encoder to generate output tokens. Unlike traditional decoders, the MERGER autoregressively predicts a sequence of $K$ tokens within a *block* of size $K$, while relying on the same precomputed contextual information from both text and audio encoders across the block. This design choice allows for the efficient reuse of encoder representations. The separation of the decoder into the text encoder and the MERGER promotes modularity and more flexible control of parameters via the number of text encoder and MERGER layers. As a result, BLOCKDECODER yields a significant speedup ($\sim 2$x) compared to traditional decoders, across diverse datasets, languages, and speech tasks, without any degradation in performance. The code is available at https://github.com/csalt-research/blockdecoder.

## 1  Introduction

Several prominent state-of-the-art automatic speech recognition (ASR) systems are derived from attention-based encoder-decoder architectures [1]. While the audio encoder's role in these architectures is to transform input speech into acoustically-rich representations, the decoder autoregressively generates text by combining acoustic information from the encoder with previously predicted text. Numerous variants of the encoder have been explored for ASR in prior work, ranging from self-supervised architectures [2, 3] to convolutionally-enriched ASR pipelines [4, 5, 6, 7]. In contrast, there have been relatively fewer innovations of the ASR decoder.

The ASR decoder comprises Transformer [8] layers with both self-attention and cross-attention modules. The self-attention module attends to previous tokens at a given time-step, and the cross-attention module attends to speech representations from the audio encoder. We analyze the behaviour of both these attention modules across all decoder layers and make the following key observations: (1) *Self-attention increasingly focuses on local context as we progress deeper into the decoder*: As shown in Figure 1, the attention weights in the initial decoder layers are well-distributed across the entire sequence.

39th Conference on Neural Information Processing Systems (NeurIPS 2025).

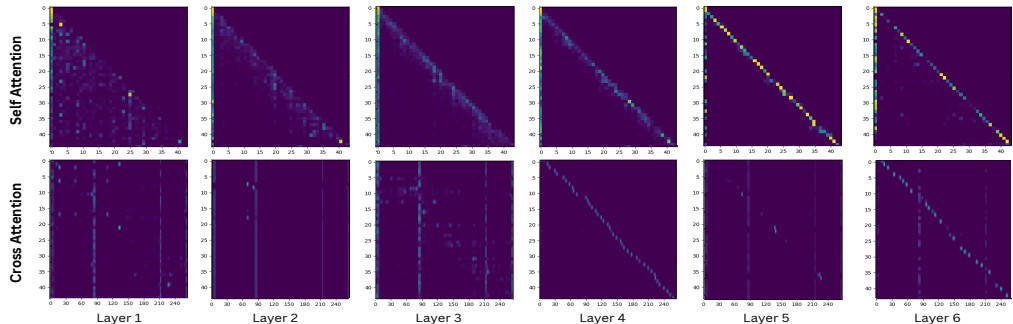

Figure 1: Attention plots illustrating the patterns learned by self-attention and cross-attention blocks across all layers of a standard Transformer decoder in a hybrid CTC/attention ASR system. The plot consists of two rows, one for self-attention and another for cross-attention and six columns, corresponding to the six decoder layers (ordered left to right). Brighter colors indicate higher attention weights. These visualizations are generated for a single example from a model trained on the Librispeech 960h dataset; we see similar patterns across most of the examples.

However, these weights become strongly diagonal as we approach the penultimate layers. A similar phenomenon has been previously observed in the audio encoder [5] but never analyzed in the decoder. (2) *Cross-attention blocks appear to be less effective in the initial decoder layers*: As illustrated in Figure 1, cross-attention in the early layers fail to learn any alignment between the audio and text sequences suggesting they might be redundant in these layers. Figure 6 in Appendix D additionally shows aggregate attention plots over 250 utterances; we see very similar patterns as illustrated in Figure 1.

Based on these systematic attention patterns, the decoder appears to be assigning specific roles to its initial and final layers. Specifically, the initial layers of the decoder are mainly tasked with the generation of text-only context by relying extensively on the token sequence, thus rendering cross-attention less effective. The final layers of the decoder merge textual context from the early layers with acoustic information from the audio encoder for its final predictions. We draw inspiration from these findings and propose BLOCKDECODER as an alternative to the conventional decoder architecture. BLOCKDECODER is partitioned into two sub-modules, a *text encoder* with only self-attention layers to build textual context and a *merger* with self-attention and two cross-attention layers to integrate acoustic representations from the audio encoder with contextualized outputs from the text encoder to generate the final predictions. Unlike traditional decoders, the merger is designed to autoregressively generate $K$ tokens (referred to as a *block* of size $K$, and hence the name BLOCKDECODER), from every position in the token sequence, while relying on the same contextual information from both encoders across the block. This block-based design enables different inference strategies, as discussed in section 3.3, and reduces inference latency. Since the merger combines representations that are already contextualized by the audio and text encoders, a small number of merger layers suffice to achieve high performance, thus resulting in additional latency reductions. Together, these complementary design choices result in BLOCKDECODER containing fewer parameters compared to the traditional decoder, while achieving double the inference speed with no performance degradation on multiple speech tasks.

## 2   Related Work

ASR systems have significantly evolved in the last decade from deep neural network-hidden Markov model (DNN-HMM) based architectures [9, 10] to fully end-to-end (E2E) ASR systems [11, 12]. These E2E models typically adopt either an encoder-only [13] or an encoder-decoder architecture [1], where the encoder processes the audio while the decoder handles text generation. Prior work has largely focused on improving these architectures either by refining individual components [14, 15] or developing joint training techniques that integrate multiple objectives to enhance model robustness [16]. One such widely-adopted jointly trained model is the CTC+attention system [17], which consists of an audio encoder, an autoregressive decoder and a CTC module. Improvements to hybrid ASR have largely focused on either making the encoder more expressive [4, 18], enhancing the robustness of CTC [19] or introducing new training objectives [20, 21]. While many efforts have aimed towards refining the audio encoder and CTC, comparatively less attention has been given to the decoder despite its critical role in text generation. BLOCKDECODER is a step towards addressing this gap.

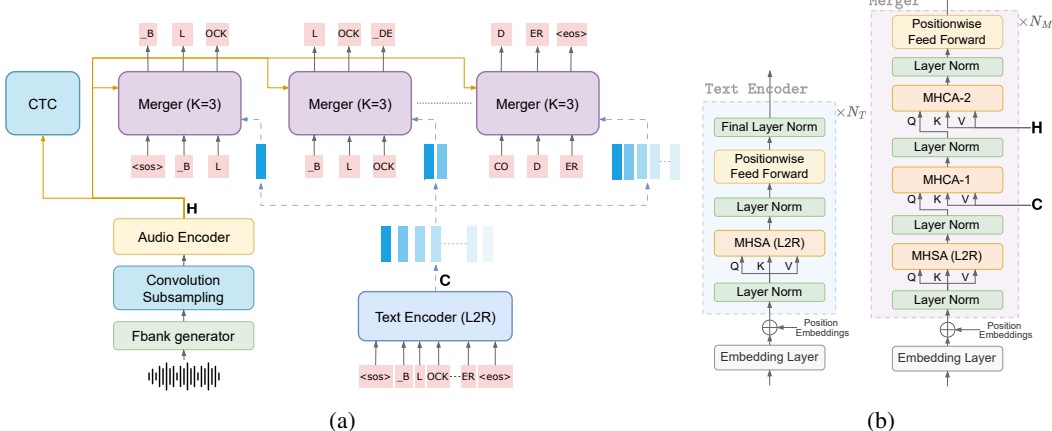

(a)                                                      (b)

Figure 2: Overview of the Hybrid CTC/Attention framework with our proposed BLOCKDECODER. (a) Schematic representation highlighting interactions between the audio encoder, text encoder, CTC, and MERGER modules during training. For the example transcript "_BLOCK_DECODER", we illustrate the MERGER's capability of predicting $K = 3$ tokens autoregressively from every position of the token sequence, while being conditioned on the same contextual representations provided by the audio encoder and text encoder. Specifically, at each stage, the MERGER receives the representation of a progressively longer text prefix (e.g., [<sos>], [<sos>, _B] etc.) from text encoder (shown via the blue bars) along with the corresponding audio representation from audio encoder (shown via orange lines), and autoregressively predicts the next three tokens. (b) A detailed expansion of the text encoder and MERGER modules, with residual connections and dropout omitted for brevity. Here, MHSA=Multi-headed Self-Attention and MHCA=Multi-headed Cross-Attention.

The autoregressive nature of the decoder in ASR systems contributes to high inference latency. To mitigate this, non-autoregressive (NAR) decoders have been explored [22, 23, 24], allowing parallel token prediction thereby improving inference efficiency. However, like CTC, NAR decoders generally underperform compared to autoregressive decoders. To bridge this gap, hybrid approaches have been proposed, where an NAR decoder first generates an initial prediction, which is then further refined by an autoregressive decoder [25]. Another line of work investigates speculative decoding techniques [26, 27], that predicts multiple future tokens in parallel, thereby further reducing inference costs. While most prior works have focused on eliminating autoregressive decoding or modifying decoding strategies [28], to the best of our knowledge, there have been no detailed investigations of the internal functioning of the decoder to refine its design. In our work, we redesign the decoder based on consistent empirical insights drawn from attention patterns. These design choices are similar in motivation to RNN-Transducer (RNN-T) systems used for streaming ASR [29], which employs separate predictor and joiner modules, each serving a distinct function. However, unlike the joiner in RNN-T, which is a simple feedforward network, we propose the MERGER, that incorporates multiple attention layers, enabling richer and more flexible context integration. Furthermore, we note that our approach operates on blocks of tokens, which is an emerging design choice in recent years [30, 31, 24].

## 3 Methodology

### 3.1 Overall Architecture

BLOCKDECODER scaffolds on top of *hybrid CTC/attention models*, a state-of-the-art framework for ASR systems [17]. This framework consists of: a shared audio encoder, an autoregressive decoder and a CTC decoder. Given an input speech sequence $\mathbf{X} = [\boldsymbol{x}_1,...,\boldsymbol{x}_{T'}], \boldsymbol{x}_i \in \mathbb{R}^d$, the audio encoder maps it to a contextualized speech representation $\mathbf{H} = [\boldsymbol{h}_1,...,\boldsymbol{h}_T].$[1] Both the CTC and autoregressive decoders are conditioned on $\mathbf{H}$ to produce the label sequence $\boldsymbol{y} = [y_1,...y_W], y_i \in \mathcal{V}$ where $\mathcal{V}$ is a predefined token vocabulary. The CTC branch minimizes the Connectionist Temporal Classification (CTC) loss [32] by marginalizing over all possible alignments of $\boldsymbol{y}$ with $\mathbf{H}$. decoder, on the other hand, is

---

[1]audio encoder typically uses subsampling to reduce the length of the speech sequence to $T = \lceil T'/S \rceil$, where $S$ is the sampling factor.

an attention-based autoregressive module that is trained to maximize the conditional likelihood of producing $y_i$ given **H** and previous tokens $y_0,...,y_{i-1}$, where $y_0$ denotes the <sos> token. In this work, we focus on the decoder by replacing it with our proposed BLOCKDECODER which is more efficient and performs at par with (or better than) the decoder.

### 3.2 BLOCKDECODER

We replace the standard decoder with two specialized modules: a text encoder and a MERGER, each designed for a specific purpose, as shown in Figure 2. The text encoder generates rich textual context for the token sequence $\boldsymbol{y}$, that the MERGER further combines along with **H**, to autoregressively generate $K$ tokens within a block. These two modules are collectively referred to as BLOCKDECODER. Fig. 2a shows a schematic overview of our ASR system incorporating these modules. If the original decoder comprises $N$ layers, then we redistribute these layers (with minor modifications) across the text encoder (in §3.2.1) and MERGER (in §3.2.2) in BLOCKDECODER to be commensurate in size with the decoder.

#### 3.2.1 Text Encoder

The text encoder is a stack of $N_T$ decoder layers that attends over the token sequence $\boldsymbol{y}$ to produce contextualized representations. These layers consist only of self-attention and feed-forward sub-layers (and no cross-attention), since text encoder operates only on the text and does not attend to the audio at all. To prevent information leakage, each token $y_i$ is allowed to attend only to itself and the preceding tokens $\{y_0,y_1,y_2,...,y_{i-1}\}$ using appropriate masking. The architecture of the text encoder is shown in Figure 2b. It consists of a stack of $N_T$ decoder layers that are designed to generate rich textual context for the token sequence $\boldsymbol{y}$ while preventing information leakage. Specifically, the output $\mathbf{C}^j$ of the $j^{\text{th}}$ text encoder layer is calculated as follows:

$$\hat{\mathbf{C}} = \text{MHSA}(\mathbf{C}^{j-1}, \mathbf{C}^{j-1}, \mathbf{C}^{j-1}, \texttt{mask})$$
$$\hat{\mathbf{C}}^j = \mathbf{C}^{j-1} + \text{LayerNorm}_{\text{att}}(\hat{\mathbf{C}})$$
$$\hat{\mathbf{C}} = \hat{\mathbf{C}}^j + \text{LayerNorm}_{\text{ff}}(\text{Linear}(\hat{\mathbf{C}}^j))$$
$$\mathbf{C}^j = \text{LayerNorm}_{\text{final}}(\hat{\mathbf{C}})$$

where $\hat{\mathbf{C}}, \mathbf{C}^j \in \mathbb{R}^{W \times d}$ and $\texttt{mask} \in \{\text{True},\text{False}\}^{W \times W}$. Here, $\text{MHSA}(*)$ denotes multi-headed self-attention [8]. During attention computation, $\texttt{mask}$ serves as a Boolean matrix, where $\texttt{mask}[a,b]$ indicates whether the $a^{\text{th}}$ token is allowed to attend to the $b^{\text{th}}$ token in $\boldsymbol{y}$. For the text encoder, we set $\texttt{mask}[a,b] = (b \leq a)$ to prevent information leakage. Finally, the input to the first layer is defined as $\mathbf{C}^0 = \text{PE} + (\text{Embedding}([y_0,y_1,...,y_W])$, where PE represents sinusoidal position embeddings. We also employ dropout [33] in every text encoder layer. Additionally, we observe that having $\text{LayerNorm}_{\text{final}}$ at the end of every layer is crucial for stable training of the BLOCKDECODER. We use $\mathbf{C} \in \mathbb{R}^{W \times d}$ to denote the final output from the text encoder.

#### 3.2.2 MERGER

Given audio context **H** from the audio encoder and text-only context **C** from the text encoder, we introduce MERGER that jointly attends to contexts from both modalities to autoregressively generate a block of $K$ tokens. MERGER achieves this integration via multiple attention blocks, as shown in Figure 2b. Specifically, MERGER first employs a self-attention block, that focuses on local context by causally attending to tokens in the current block. Next, MERGER uses two cross-attention modules for **C** and **H**, respectively. The first cross-attention module attends to the contextualized representations in **C** prior to the current block; the second cross-attention block integrates acoustic features from **H**, thereby ensuring that the tokens generated by MERGER remain faithful to the audio. At the $i^{\text{th}}$ time step, MERGER is trained to maximize the probability of generating the token sequence $\{y_i,...y_{i+K-1}\}$, conditioned on **H**, **C** and the previous token $y_{i-1}$:

$$P(\{y_i, ... y_{i+K-1}\} \mid y_{i-1}, \mathbf{H}, \mathbf{C}_{0:i-1}) = \prod_{k=0}^{K-1} \text{MERGER}(\{y_{i-1}, ... y_{i+k-1}\}, \mathbf{H}, \mathbf{C}_{0:i-1})$$

It is important to note that at time-step $i$, while generating each of $K$ tokens in a block, MERGER conditions on the same first $i$ entries of **C** written as $\mathbf{C}_{0:i-1}$. Since MERGER can predict a block of

$K$ future tokens at each time step, we note that the probability for any token in the block can now be computed using multiple combinations of entries from **C** and the set of preceding tokens. For example, with a block size of 3, the probability of $y_5$ can be computed using the following combinations: $(\{y_4\},\mathbf{C}_{0:4})$, $(\{y_3,y_4\},\mathbf{C}_{0:3})$ and $(\{y_2,y_3,y_4\},\mathbf{C}_{0:2})$. Section 3.3 elaborates on multiple inference strategies supported by BLOCKDECODER.

To facilitate efficient attention and loss computations, we create a modified token sequence $\boldsymbol{y}' = [y_0,y_1,...,y_{K-1},y_1,y_2,...,y_K,...,y_{W-K+1},...,y_W]$ of length $|\boldsymbol{y}'| = K \times (W-K+1)$. This sequence contains $(W-K+1)$ blocks, each comprising $K$ tokens, that are color-coded for improved readability. (Since the length of $\boldsymbol{y}'$ grows linearly with $K$, we typically employ small values for $K$.) This modified sequence now enables us to restrict self- and cross-attention by appropriately configuring the attention masks. Specifically, the output of the $j^{\text{th}}$ layer $\mathbf{M}^j$ from MERGER (consisting of $N_M$ layers overall) is computed as:

$$\hat{\mathbf{M}}^j = \mathrm{MHSA}(\mathbf{M}^{j-1},\mathbf{M}^{j-1},\mathbf{M}^{j-1},\mathtt{mask}_{\text{self}})$$

$$\hat{\mathbf{M}} = \mathbf{M}^{j-1} + \mathrm{LayerNorm}_{\text{self}}(\hat{\mathbf{M}}^j)$$

$$\hat{\mathbf{M}}^j = \mathrm{MHCA}_{\text{context}}(\hat{\mathbf{M}},\mathbf{C},\mathbf{C},\mathtt{mask}_{\text{context}})$$

$$\hat{\mathbf{M}} = \hat{\mathbf{M}} + \mathrm{LayerNorm}_{\text{context}}(\hat{\mathbf{M}}^j)$$

$$\hat{\mathbf{M}}^j = \hat{\mathbf{M}} + \mathrm{LayerNorm}_{\text{audio}}(\mathrm{MHCA}_{\text{audio}}(\hat{\mathbf{M}},\mathbf{H},\mathbf{H}))$$

$$\mathbf{M}^j = \hat{\mathbf{M}} + \mathrm{LayerNorm}_{\text{ff}}(\mathrm{Linear}(\hat{\mathbf{M}}^j)$$

where $\hat{\mathbf{M}}^*,\mathbf{M}^j \in \mathbb{R}^{W \times d}$, $\mathrm{MHSA}(*)$ and $\mathrm{MHCA}(*)$ denote the standard multi-headed self- and cross-attention [8], and $\mathtt{mask}_{\text{self}}, \mathtt{mask}_{\text{context}} \in \{0,1\}^{W \times W}$ are Boolean matrices where $\mathtt{mask}_*[a,b] = 1$ indicates that the $a^{\text{th}}$ item in one sequence is allowed to attend to the $b^{\text{th}}$ item in another sequence. To constrain self-attention to causally attend over tokens within the block, we set $\mathtt{mask}_{\text{self}}[a,b] = (b < a) \,\&\, (\lfloor \frac{b}{K} \rfloor == \lfloor \frac{a}{K} \rfloor)$. To prevent information leakage from **C**, we set $\mathtt{mask}_{\text{context}}[a,b] = b \leq \lfloor \frac{a}{K} \rfloor$. Lastly, we use embeddings for tokens within each block of the sequence $\boldsymbol{y}'$ with appropriate positional embeddings as an input to MERGER. Finally, the output probabilities from MERGER are generated using a simple feed-forward layer to project to the token vocabulary size $\mathcal{V}$, followed by a softmax transformation.

To ensure that **C** is contextually rich, we allocate the majority of our layer budget to the text encoder. With **H** and **C** being substantially rich in information, even a small number of MERGER layers are adequate to effectively combine both modalities. This careful division of responsibilities between a cross-attention-free text encoder (that generates contextualized textual representations) and a lightweight MERGER (that allows late integration of speech and text information) enables BLOCKDECODER to function accurately and significantly faster than the traditional decoder.

### 3.3 Inference Strategies

During inference, we employ label-synchronous autoregressive beam search proposed by [17] to find the best hypothesis. That is, at the $i^{\text{th}}$ decoding step, the score for the partially decoded hypothesis $\hat{\boldsymbol{y}}_{\leq i}$ is computed as a weighted combination of the log probabilities from both CTC and BLOCKDECODER. Specifically:

$$\mathcal{S}(\hat{\boldsymbol{y}}_{\leq i}) = \delta \times \mathcal{S}_{\text{ctc}}(\hat{\boldsymbol{y}}_{\leq i}) + (1-\delta) \times \mathcal{S}_{\text{att}}(\hat{\boldsymbol{y}}_{\leq i}) \tag{1}$$

where $\delta \in [0,1]$ is a hyperparameter that determines the relative importance of the two scores $\mathcal{S}_{\text{ctc}}$ and $\mathcal{S}_{\text{att}}$ from the CTC and BLOCKDECODER modules, respectively. $\mathcal{S}_{\text{att}}$ can be computed using various combinations of inputs to the text encoder and MERGER. We explore three inference strategies to compute the score for the partially decoded hypothesis $\hat{\boldsymbol{y}}_{\leq i}$, as illustrated in Figure 3.

#### 3.3.1 Strategy 1: Naive Block Decoding

The simplest approach involves making a forward pass through both text encoder and MERGER at every decoding step, by utilizing all positions within the block. Specifically, the attention score for $\hat{\boldsymbol{y}}_{\leq i}$ is computed as:

$$\mathcal{S}_{\text{att}}(\hat{\boldsymbol{y}}_{\leq i}) = \log(\prod_{j=1}^{i} \mathrm{MERGER}(\{y_{j-K},...,y_{j-1}\},\mathbf{H},\mathbf{C}'))$$

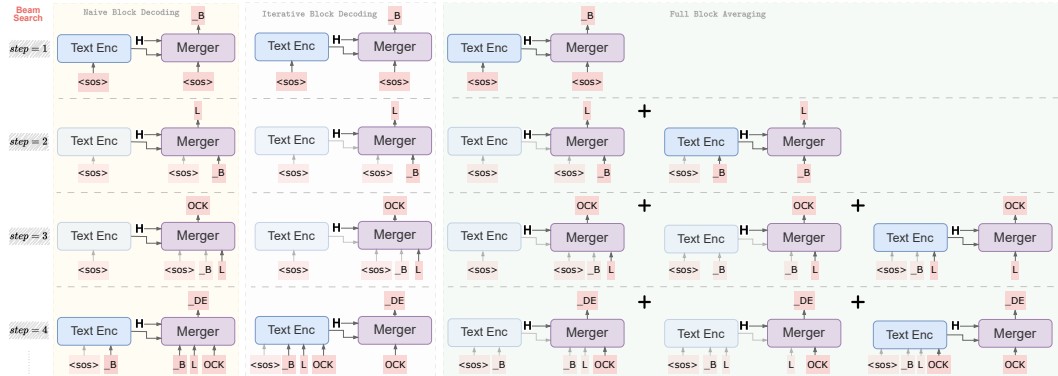

Figure 3: Overview of the three inference strategies across the first four beam search steps. "Text Enc" refers to the text encoder. From left to right: (1) **Naive Block Decoding**: Score obtained from only the last position within the block, requiring one forward pass through both the text encoder and MERGER at each step. (2) **Iterative Block Decoding**: Score computed iteratively within the block, requiring one forward pass through MERGER at each step, and one pass through text encoder every $K$ steps. (3) **Full Block Averaging**: Score averaged across all valid input combinations to text encoder and MERGER, requiring one forward pass through text encoder and multiple passes through MERGER at each step.

where $\mathbf{C}' = \text{text encoder}(\{y_0, y_1, ... y_{j-K}\})$. This approach always utilizes the last position within the block while generating the score. This approach is sub-optimal as it involves forward passes through both text encoder and MERGER at every decoding step.

### 3.3.2 Strategy 2: Iterative Block Decoding

To truly exploit the MERGER's blockwise ability to predict $K$ tokens, we adopt an *iterative block decoding* approach, where the MERGER continuously predicts $K$ tokens before performing a forward pass through the text encoder to update the context vector $\mathbf{C}'$. Here, the score for $\hat{\boldsymbol{y}}_{\leq i}$ is computed as:

$$\mathcal{S}_{\text{att}}(\hat{\boldsymbol{y}}_{\leq i}) = \log\left(\prod_{j=1}^{i} \text{MERGER}(\{y_s, ..., y_{j-1}\}, \mathbf{H}, \mathbf{C}')\right)$$

where $\mathbf{C}' = \text{text encoder}(\{y_0, y_1, ... y_s\})$ and $s = \lfloor \frac{j-1}{K} \rfloor \times K$. Since the MERGER contains only a few layers and we make a forward pass through text encoder once every $K$ steps, this strategy is more efficient compared to the previous strategy.

### 3.3.3 Strategy 3: Full Block Averaging

This last approach aims to boost performance by utilizing the scores from all the positions within the block. The final score for $\hat{\boldsymbol{y}}_{\leq i}$ is computed as the average of the scores from all the combinations of $\mathbf{C}'$ and the preceding tokens as shown below:

$$\mathcal{S}_{\text{att}}(\hat{\boldsymbol{y}}_{\leq i}) = \log\left(\prod_{j=1}^{i} \frac{1}{K} \sum_{k=1}^{K} \text{MERGER}(\{y_{j-k}, .., y_{j-1}\}, \mathbf{H}, \mathbf{C}')\right)$$

where $\mathbf{C}' = \text{text encoder}(\{y_0, y_1, ... y_{j-k}\})$. While this strategy improves performance, it incurs higher inference latency due to $K$ invocations of MERGER as opposed to a single invocation in the previous strategy.

### 3.4 Complexity of BLOCKDECODER

We analyze the computational complexity of BLOCKDECODER in comparison to the standard decoder. Let $T$ be the length of the acoustic features $\mathbf{H}$, $W$ denote the length of the token sequence $\boldsymbol{y}$, $K$ be the block size for the MERGER and $d$ denote the dimensionality for the attention computations. (For ASR, typically $W \ll T$.) BLOCKDECODER consists of $N_T$ and $N_M$ layers in text encoder and MERGER, respectively. (Recall that $N = N_T + N_M$.)

Table 1: Performance comparisons (CER or WER %) between BLOCKDECODER and the baseline Transformer decoder across three datasets and two encoder architectures. ▨ indicates no statistically significant WER difference compared to the baseline, as determined by the MAPSSWE test [34]. ▨ denotes significant improvement in RTF, while ▷ indicates RTF performance comparable to the baseline.

| Method | Librispeech-100h (WER) | | | | Tedlium2 (WER) | | | AISHELL (CER) | | |
|---|---|---|---|---|---|---|---|---|---|---|
| | Params (M) | Test Clean ↓ | Test Other ↓ | RTF ↓ | Params (M) | Test ↓ | RTF ↓ | Params (M) | Test ↓ | RTF ↓ |
| Conformer [4] | | | | | | | | | | |
| w/ Transformer Decoder | 34.2 | 6.75 | 17.74 | 1.38 | 30.8 | **7.69** | 2.16 | 33.6 | **4.58** | 0.44 |
| w/ BLOCKDECODER | | | | | | | | | | |
| − Naive Block Decoding | 33.7 | 6.63 | 17.63 | 0.73$_{(\sim1.9x)}$ | 30.2 | 7.81 | 1.02$_{(\sim2.1x)}$ | 33.1 | 4.76 | 0.28$_{(\sim1.6x)}$ |
| − Iterative Block Decoding | 33.7 | 6.72 | 17.70 | **0.66**$_{(\sim2.1x)}$ | 30.2 | 7.92 | **0.90**$_{(\sim2.4x)}$ | 33.1 | 4.75 | **0.25**$_{(\sim1.8x)}$ |
| − Full Block Averaging | 33.7 | **6.58** | **17.62** | 1.68 ▷ | 30.2 | 7.79 | 2.39 ▷ | 33.1 | 4.63 | 0.51 ▷ |
| E-Branchformer [6] | | | | | | | | | | |
| w/ Transformer Decoder | 38.5 | 6.39 | 17.03 | 1.52 | 35.0 | **7.44** | 2.17 | 37.9 | **4.50** | 0.45 |
| w/ BLOCKDECODER | | | | | | | | | | |
| − Naive Block Decoding | 37.9 | 6.15 | **16.82** | 0.77$_{(\sim2.0x)}$ | 34.5 | 7.61 | 1.04$_{(\sim2.1x)}$ | 37.4 | 4.61 | 0.30$_{(\sim1.5x)}$ |
| − Iterative Block Decoding | 37.9 | 6.19 | 16.94 | **0.67**$_{(\sim2.3x)}$ | 34.5 | 7.60 | **0.91**$_{(\sim2.4x)}$ | 37.4 | 4.61 | **0.26**$_{(\sim1.7x)}$ |
| − Full Block Averaging | 37.9 | **6.14** | 16.85 | 1.68 ▷ | 34.5 | 7.61 | 2.40 ▷ | 37.4 | 4.53 | 0.54 ▷ |

We focus primarily on comparing the total number of attention calculations required for a single example during training and inference.

During training, the text encoder takes $y$ as its input while the MERGER uses the modified sequence $y'$ as its input, where $|y'| = (W - K + 1)K \approx WK$. Thus, the overall attention computation for BLOCKDECODER has a complexity of $O(N_T(W^2d) + N_M(WK^2 + W^2K + WKT)d)$ in comparison to decoder that takes $O(N(W^2 + WT)d)$ time. Upon further simplification (detailed in Appendix A), we show that choosing $K \approx \frac{N}{2N_M}$ allows BLOCKDECODER to operate with approximately the same number of attention operations as decoder.

During inference with beam-search (of $B$ beam width), the standard decoder takes $O(WB(NW + NT)d)$ time for its attention calculations. In contrast, BLOCKDECODER with naive block decoding takes $O(WB(NW + N_M(K+T))d)$ time. Since $K \ll T$ and $N_M \ll N$, BLOCKDECODER requires significantly fewer attention operations than decoder. The time complexity of iterative block decoding is similar to that of the naive strategy, but benefits from making a forward pass through text encoder only once every $K$ steps thus leading to slightly faster inference. The complexity of full block averaging and other details are in Appendix A.

Table 2: WERs of BLOCKDECODER and other well-known ASR systems (from prior work) on Librispeech 960h.

| Method | Params (M) | Without LM | | |
|---|---|---|---|---|
| | | Test Clean ↓ | Test Other ↓ | RTF ↓ |
| **Transducer** | | | | |
| Transformer [21] | 139 | 2.4 | 5.6 | - |
| ContextNet [14] | 112.7 | **2.1** | 4.6 | - |
| Conformer (M) [4] | 30.7 | 2.3 | 5.0 | - |
| Conformer (L) [4] | 118.8 | **2.1** | 4.3 | - |
| **CTC** | | | | |
| QuartzNet (L) [35] | 19 | 3.9 | 11.3 | - |
| **Hybrid CTC+Attention** | | | | |
| Transformer [36] | 270 | 2.9 | 7.0 | - |
| Conformer [37] | 116.2 | 2.9 | 7.0 | - |
| Conformer (L) [6] | 147.8 | 2.2 | 4.7 | - |
| Branchformer [5] | 116.2 | 2.4 | 5.5 | - |
| Branchformer (L) [6] | 146.7 | 2.2 | 4.8 | - |
| E-Branchformer [6] | 148.9 | **2.1** | **4.5** | - |
| **Our baselines (Hybrid)** | | | | |
| E-Branchformer | | | | |
| − w/ Transformer Decoder | 148.9 | **2.1** | 4.6 | 7.734 |
| **Our work (Hybrid)** | | | | |
| − w/ BLOCKDECODER | 146.8 | **2.1** | **4.4** | 2.983$_{(\sim2.6x)}$ |

Table 3: Comparison (accuracy % and F1) of our system against Transformer decoder baseline on the SLU task.

| Method | Params (M) | Intent Test Acc. ↑ | Entity SLU-F1 ↑ | RTF ↓ |
|---|---|---|---|---|
| Conformer [4] | | | | |
| − w/ Transf. Decoder | 109.5 | **85.9** | 0.76 | 2.46 |
| − w/ BLOCKDECODER | 107.4 | **85.9** | 0.77 | **1.17**$_{(\sim2.1x)}$ |
| E-Branchformer [6] | | | | |
| − w/ Transf. Decoder | 110.2 | **87.2** | 0.78 | 3.11 |
| − w/ BLOCKDECODER | 108.1 | 87.1 | 0.78 | **1.31**$_{(\sim2.4x)}$ |

## 4 Experiments

### 4.1 Experimental Setup

**Datasets.** We show experiments on two tasks: ASR and Spoken Language Understanding (SLU). For ASR, we use: (1) Librispeech [38] consisting of 1000 hours of English read audiobooks with 100-hour and 960-hour training splits, (2) Tedlium2 [39] consisting of 200 hours of TED talk recordings,

Table 4: CERs/WERs of BLOCKDECODER and the baseline decoder across five languages from the MCV corpus. All experiments use the E-Branchformer as encoder. ◇ indicates the languages for which CER is reported.

| Language | Decoder used | Params | Dev ↓ | Test ↓ | RTF ↓ |
|---|---|---|---|---|---|
| **Chinese** ◇ | Transformer | 51.1M | 14.0 | 13.6 | 0.41 |
| | Ours | 50.5M | 14.2 | 13.8 | **0.22**$_{(\sim1.9x)}$ |
| **Czech** | Transformer | 47.4M | 12.4 | 13.4 | 0.49 |
| | Ours | 46.9M | 12.7 | 13.8 | **0.25**$_{(\sim2.0x)}$ |
| **Italian** | Transformer | 47.4M | 9.5 | 10.1 | 0.73 |
| | Ours | 46.9M | 9.7 | 10.5 | **0.37**$_{(\sim2.0x)}$ |
| **Japanese** ◇ | Transformer | 49.9M | 5.3 | 12.1 | 0.43 |
| | Ours | 49.4M | 5.5 | 12.3 | **0.28**$_{(\sim1.5x)}$ |
| **Tamil** | Transformer | 47.4M | 17.4 | 20.0 | 0.57 |
| | Ours | 46.9M | 17.5 | 20.1 | **0.31**$_{(\sim1.8x)}$ |

Table 5: WERs of BLOCKDECODER with varying block sizes on Librispeech 100h using two training strategies: (1) **full**: MERGER trained on every block (2) **sampled**: MERGER trained on a random subset of blocks.

| Block size | Training Strategy | Training Time (hrs) | Test Clean ↓ | Test Other ↓ | RTF ↓ |
|---|---|---|---|---|---|
| **Baseline** | – | 15.4 | 6.4 | 17.0 | 1.52 |
| $K=1$ | full | 15.2 | 6.7 | 17.3 | 0.73$_{(\sim2.1x)}$ |
| $K=3$ | full | 15.6 | 6.2 | 16.9 | 0.67$_{(\sim2.3x)}$ |
| $K=5$ | full | 16.2 | 6.3 | 17.1 | 0.66$_{(\sim2.3x)}$ |
| | sampled | 15.4 | 6.4 | 17.7 | 0.65$_{(\sim2.3x)}$ |
| $K=7$ | full | 17.0 | 6.2 | 16.9 | 0.65$_{(\sim2.3x)}$ |
| | sampled | 15.3 | 6.9 | 18.2 | 0.65$_{(\sim2.3x)}$ |
| $K=9$ | full | 17.5 | 6.5 | 17.4 | 0.65$_{(\sim2.3x)}$ |
| | sampled | 15.3 | 14.9 | 29.6 | 0.65$_{(\sim2.3x)}$ |

(3) Aishell [40] containing 170 hours of Mandarin Chinese speech data, and (4) Mozilla Common-Voice [41], a multilingual dataset with durations ranging from 10 to 2500 hours per language. We select five languages with training data spanning 100 to 400 hours. For SLU, we use the SLURP corpus [42], a 60-hour multi-domain English dataset evaluated for intent classification and entity recognition.

**Implementation Details.** All our experiments are conducted using the ESPnet toolkit [43] on NVIDIA A100 and A6000 GPUs.[2] Across all experiments, we apply 3-way speed perturbation with ratios $\{0.9, 1.0, 1.1\}$, along with SpecAugment [44]. Our experimental setup follows the recommended configurations in ESPnet recipes. Across all experiments, we employ standard efficient inference techniques such as KV-caching and Automatic Mixed Precision (AMP). For SLU experiments, we first train the model with an ASR objective, where the output label sequences are sentences with intent and entity-related tags. Then, during inference, we first decode the sequence as in ASR, and then compute SLU metrics by parsing the decoded output. A detailed summary of the hyperparameters used for each dataset is available in Appendix F. Real-time factor (RTF) [3] values are reported using CPU inference for all experiments. Finally, unless explicitly stated, all BLOCKDECODER related experiments use a block size of $K=3$, $N_T=4$ text encoder layers, and $N_M=2$ MERGER layers.

## 4.2 Main Results

Table 1 compares the word error rates (WERs) of BLOCKDECODER with a standard Transformer decoder (referred to as baseline) across three widely used ASR tasks: Librispeech 100h and Tedlium2 for English, and Aishell for Mandarin. For each dataset, we experiment with two state-of-the-art encoder architectures: Conformer [4] and E-Branchformer [6] and report results on all three inference strategies outlined in Section 3.3. We find that BLOCKDECODER consistently matches or outperforms the baselines despite having fewer parameters and achieves significant latency gains. The first two inference strategies are fast, achieving a near 2x speedup in RTF by effectively utilizing the block structure while the third strategy does not improve RTF but yields a slight performance boost. Iterative block decoding yields the best RTF gains and will be used in all subsequent experiments (unless specified otherwise).

In Table 2, we further compare the WERs between BLOCKDECODER and decoder on the full Librispeech 960-hour dataset, along with other prominent baselines from prior work. Even on a large-scale dataset, BLOCKDECODER remains comparable in performance with the strongest baseline, E-Branchformer, while achieving nearly a 2.5x speedup in RTF on CPUs. Additionally, since inference on Librispeech 960h often employs a large beam size, in Appendix E, we also compare the performance of BLOCKDECODER against the decoder with inference using GPUs. Even in this setting, BLOCKDE-CODER achieves better RTF and FLOPs (Floating Point Operations) compared to the standard decoder.

## 4.3 More Task and Language Experiments

**Non-English ASR.** In Table 4, we show additional experiments on five languages from diverse language families of the MCV corpus (Chinese, Czech, Italian, Japanese, Tamil). BLOCKDECODER

---

[2]We ensure that all experiments for a particular dataset are conducted on the same GPU and environment.
[3]RTF is the ratio between the time taken by the model to process the input and the actual input duration.

consistently yields WERs comparable to the baseline across all languages while maintaining significant latency gains.

**SLU task.**  In Table 3, we show results on an SLU task, SLURP [42], and evaluate using accuracy and F1-score on intent classification and entity recognition, respectively. As with ASR, BLOCKDECODER for SLU also results in comparable performance to the baseline and significantly improves latency.

## 4.4  Ablations and Analysis

**Attention plots.**  Figure 4 shows the attention patterns learned by the text encoder and MERGER trained on Librispeech 960h. Figure 4a shows self-attention plots of the text encoder with well-distributed attention distributions across the sequence for all layers, indicating that the text encoder focuses on a global, text-only context.  Figures 4b and 4c show attention plots for the two cross-attention blocks of the MERGER.  The first cross-attention block that attends to text encoder's output is predominantly diagonal, reinforcing the idea that the global context produced by the text encoder is sufficiently informative for the MERGER.

The second cross-attention block, that attends to the audio encoder's output, demonstrates accurate alignment with the audio features, thus highlighting the importance of these blocks in integrating contexts.

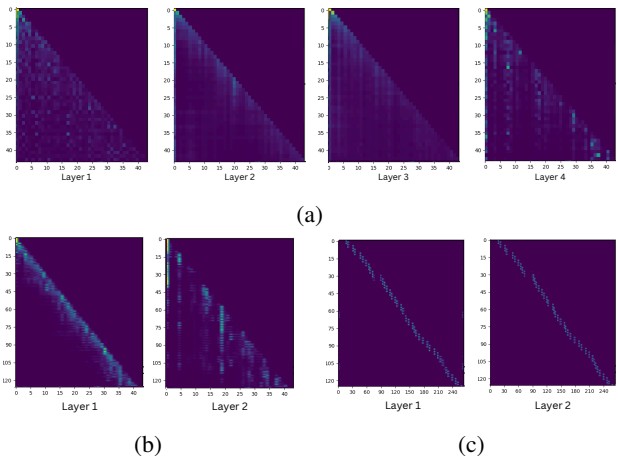

(a)

(b)          (c)

Figure 4:  Attention plots demonstrating the patterns learned by self- and cross-attentions from BLOCKDE-CODER.  (a) text encoder's self-attention (b) MERGER's cross-attention w/ text encoder (c) MERGER's cross-attention w/ audio encoder.

**Impact of the number of MERGER layers.**  Figure 5 illustrates the effect of varying the number of MERGER layers ($N_M$) in the BLOCKDECODER. We see that a single MERGER layer (i.e. $N_M = 1$) is sufficient to achieve reasonable WERs while yielding a significant latency gain with a nearly 3.5x speedup.  Adding a second layer (i.e $N_M = 2$) matches the baseline in WER, while still maintaining a 2x latency gain. As the number of MERGER layers increases, performance continues to improve, while consistently achieving better RTF than the baseline.

**Impact of block size K.**  Table 5 presents a comparison of BLOCKDE-CODER's performance on Librispeech 100h for varying block sizes $K = \{1, 3, 5, 7, 9\}$. As larger $K$ increases training time, we also explore an alternative training strategy in which the MERGER is trained on only a random subset of $\frac{|y|}{K}$ blocks such that the total number of tokens is roughly equivalent in length to the original input $y$ (marked as *sampled*).  All non-sampled variants achieve performance comparable to the baseline

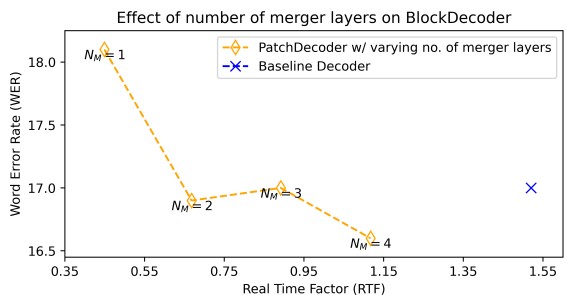

Figure 5:  Comparison of the performance of BLOCKDE-CODER on changing number of MERGER layers. WERs are from the test-other split of Librispeech 100h.

while demonstrating substantial RTF improvements. RTF gains initially increase with $K$ since iterative block decoding requires fewer forward passes through text encoder. The gains subsequently plateau with larger $K$ values due to the increased computational overhead of the MERGER that offsets latency benefits gained from skipping text encoder. Additionally, as $K$ increases, we see a slight degradation in performance, due to the increased complexity of the MERGER's learning objective. All sampling-based

experiments underperform relative to their non-sampled counterparts, likely due to insufficient training, but match the baseline in terms of training time.

**BLOCKDECODER versus CTC only models.** CTC-only models are a popular alternative to encoder-decoder architectures due to their fast inference enabled by greedy decoding. However, they are known to significantly underperform compared to encoder-decoder models. Hybrid CTC encoder-decoder architectures are a popular choice to achieve state-of-the-art performance; hence, we primarily focus on this framework. Further detailed performance comparison between BLOCKDECODER and multiple CTC-only models (that focus on efficiency) are presented in Appendix C.

**Decoder without cross-attention layers.** We also investigate a simplified variant of BLOCKDE-CODER, which retains the decoder architecture but removes cross-attention blocks from the initial layers. This modification results in worse performance compared to our proposed BLOCKDECODER. Further details are in Appendix B.

**Adaptation of BLOCKDECODER to other settings.** Our proposed architecture is particularly effective in settings where the encoder output is substantially longer than the target sequence (e.g., ASR). Similar input-output dynamics exist in tasks such as document summarization, which often involves long input sequences and shorter outputs, making them promising candidates for adaptations of BLOCKDECODER. Moreover, since BLOCKDECODER separates the decoder into a text-only encoder and a merger module, our framework naturally supports replacing the text encoder with a pretrained LLM, thereby enabling seamless ASR-LLM integration. Finally, the architectural modularity we introduce could inspire efficient pruning or compression strategies for large-scale models such as Whisper [1], thereby accelerating inference, especially for real-time applications.

## 5 Conclusion

In this work, we propose BLOCKDECODER, a novel decoder architecture for ASR inspired by attention patterns observed in traditional Transformer decoders, specifically that cross-attention is under-utilized in initial layers and self-attention becomes more localized in later layers. BLOCKDECODER consists of a text encoder, tasked with generating rich text-only context and a MERGER, responsible for efficiently integrating audio and text contexts to autoregressively produce $K$ tokens within a block. Together, these modules ensure that BLOCKDECODER is on-par with (or better than) the standard decoder, while achieving significant RTF gains across a wide range of languages and tasks. The block structure within the MERGER opens up avenues for further investigations into inference methods, optimizations, and additional performance enhancements.

## Acknowledgments

The authors would like to thank all the anonymous reviewers and chairs for their valuable feedback and constructive suggestions, resulting in significant improvements in the quality of this submission.

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

# A  Detailed Complexity Analysis of BLOCKDECODER

Let $T$ denote the length of the acoustic features $\mathbf{H}$, $W$ be the length of the token sequence $\boldsymbol{y}$, $K$ be the block size for the MERGER and let $d$ be the dimensionality for the attention computations. Additionally, $N$ is the total number of layers in the original decoder, and $N_T$ and $N_M$ are the number of layers in text encoder and MERGER respectively. As discussed in Section 3.2, we choose $N_T$ and $N_M$ such that $N = N_T + N_M$. The goal here, as outlined in Section 3.4, is to mainly compare the total number of attention computations between our BLOCKDECODER and the traditional decoder. To begin with, we know that, during training, decoder takes $O(NW^2d + NWTd)$ time for all its attention calculations. In comparison, BLOCKDECODER has a time complexity of:

$$O(\underbrace{N_T(W^2d)}_{\text{text encoder}} + N_M(\underbrace{WK^2d}_{\substack{\text{Self-Attention} \\ \text{within the block}}} + \underbrace{W^2Kd}_{\substack{\text{Cross-Attention} \\ \text{over } \mathbf{C}}} + \underbrace{WKTd}_{\substack{\text{Cross-Attention} \\ \text{over } \mathbf{H}}}))$$

$$\underbrace{\phantom{WK^2d + W^2Kd + WKTd}}_{\text{MERGER}}$$

Simplifying this expression, we get:

$$
\begin{aligned}
&= O(N_T(W^2d) + N_M(WK^2d + W^2Kd + WKTd)) \\
&= O((N - N_M)(W^2d) + N_M(WK^2d + W^2Kd + WKTd)) \quad \text{(Since } N = N_T + N_M\text{)} \\
&= O(N(W^2d) + N_M(WK^2d + W^2Kd - W^2d + WKTd)) \\
&\approx O(NW^2d + N_M(WK^2d + W^2Kd + WKTd)) \\
&\approx O(NW^2d + N_M WKd(K + W + T)) \\
&\approx O(NW^2d + N_M WKd(2T)) \quad \text{(Since } K \ll T \text{ and } W \ll T\text{)} \\
&\approx O(NW^2d + 2N_M KWTd))
\end{aligned}
$$

Doing an elementwise comparison with the time complexity of the decoder, we find:

$$N \approx 2N_M K \implies K \approx \frac{N}{2N_M}$$

Similarly, during inference, we employ beam search to generate the $B$-best hypothesis, where $B$ denotes the beam width. In the case of the traditional decoder, the time complexity for performing inference on $\boldsymbol{y}$ is $O(WBN(W + T)d)$. That is, in total, we perform $W$ beam steps (since $|\boldsymbol{y}| = W$), with each step processing $B$ partial hypothesis, which are all passed through $N$ decoder layers, where each layer first performs self-attention over the previously generated tokens, followed by cross-attention over the acoustic features $\mathbf{H}$. These operations result in $W + T$ attention computations, for a single forward pass of one entry in the beam. In contrast, we show that our BLOCKDECODER significantly reduces the number of attention computations required for beam-search. Given that BLOCKDECODER supports multiple inference strategies, we now analyze the time complexity for each of those approaches.

**NAIVE BLOCK DECODING & ITERATIVE BLOCK DECODING:**  Although ITERATIVE BLOCK DECODING makes a forward pass through text encoder once every $K$ steps, asymptotically, the total number of attention computations are still equivalent to NAIVE BLOCK DECODING. Thus, to calculate the overall time complexity, we assume that a forward pass through both the text encoder and MERGER at each beam-step is performed. Then, the time complexity becomes:

$$O(WB(N_T W + N_M(K + W + T))d)$$

Simplifying this expression, we get:

$$
\begin{aligned}
&= O(WB(N_T W + N_M(K + W + T))d) \\
&= O(WB((N_T + N_M)W + N_M(K + T))d) \\
&= O(WB(NW + N_M(K + T))d) \quad \text{(Since } N = N_T + N_M\text{)} \\
&\approx O(WB(NW + N_M T)d) \quad \text{(Since } K \ll T\text{)}
\end{aligned}
$$

Finally, since $N_M \ll N$, the overall time complexity is significantly smaller than that of the standard decoder. Although ITERATIVE BLOCK DECODING is of the same time complexity as NAIVE BLOCK DECODING, since modern hardwares are optimized for parallel computations, it further reduces inference time.

FULL BLOCK AVERAGING: This strategy involves making multiple forward passes through the MERGER. As a result, it has an overall time complexity of:

$$O(WB(N_T W + N_M K(K+W+T))d)$$

Simplifying this expression, we get:

$$
\begin{aligned}
&= O(WB(N_T W + N_M K(K+W+T))d) \\
&= O(WB((N-N_M)W + N_M K(K+W+T))d) \quad \text{(Since } N = N_T + N_M\text{)} \\
&= O(WB(NW + N_M(K^2 + (K-1)W + KT))d) \\
&\approx O(WB(NW + N_M(K^2 + KW + KT))d) \\
&\approx O(WB(NW + N_M K(K+W+T))d) \\
&\approx O(WB(NW + N_M \frac{N}{2N_M}(K+W+T))d) \quad \text{(Since } K \approx \frac{N}{2N_M}\text{)} \\
&\approx O(WB(NW + \frac{N}{2}(K+W+T))d) \\
&\approx O(WB(NW + \frac{N}{2}(W+T))d) \quad \text{(Since } K \ll T\text{)} \\
&\approx O(WB(\frac{3NW}{2} + \frac{NT}{2}))d) \\
&\approx O(WBN(W+T))d)
\end{aligned}
$$

Thus, with $K \approx \frac{N}{2N_M}$, this strategy requires approximately similar number of operations as the standard decoder.

## B  BLOCKDECODER vs decoder with cross-attentions only in the later layers

A simpler variant of our architecture can be designed by removing cross-attention blocks from few initial layers of the standard decoder. Since cross-attention in the decoder typically causes the performance bottleneck, reducing the number of such blocks should improve the decoder efficiency. Table 6 presents a performance comparison between the baseline, this simplified decoder variant and our BLOCKDECODER. We observe that removing cross-attention blocks does provide significant latency gains, but at the cost of performance degradation. However, our proposed BLOCKDECODER still achieves the best RTF gain and exhibits no performance degradation, owing to its ability to effectively utilize the block structure.

Table 6: Performance comparison (CER or WER %) between Transformer decoder, our BLOCK-DECODER and a variant of decoder containing cross-attentions only at the later layers (referred to as decoder_efficient) on Librispeech 100h dataset. All experiments use the E-Branchformer as encoder.

| Architecture | Params | Test Clean ↓ | Test Other ↓ | RTF ↓ |
|---|---|---|---|---|
| Transformer decoder | 38.5M | 6.39 | 17.03 | 1.52 |
| decoder_efficient | 37.5M | 6.51 | 17.26 | $\mathbf{0.73}_{(\sim 2.1\text{x})}$ |
| BLOCKDECODER | 37.9M | 6.19 | 16.94 | $\mathbf{0.66}_{(\sim 2.3\text{x})}$ |

## C  BLOCKDECODER vs CTC-only architectures

In this section, we evaluate BLOCKDECODER against vanilla CTC-only models and a widely used CTC-only variant, InterCTC [45], which introduces auxiliary CTC objectives at intermediate encoder layers.

Since CTC-only models rely solely on the encoder, we increase their number of encoder layers to match the total number of layers (encoder + decoder) in BLOCKDECODER. Table 7 presents a performance comparison between the standard Transformer Decoder, the vanilla CTC-only model, InterCTC, and our BLOCKDECODER. As shown, CTC-only models yield significantly worse performance compared to both the full Transformer decoder and BLOCKDECODER. Although CTC-only models offer faster inference due to their greedy decoding, this comes at a notable cost in recognition accuracy. These results support our claim that BLOCKDECODER achieves a more favorable trade-off between latency and performance.

Table 7: Performance comparison (CER or WER %) between Transformer decoder, vanilla CTC-only, InterCTC, and our BLOCKDECODER on Librispeech 100h dataset. All experiments use the E-Branchformer as an encoder.

| Architecture | Params | Test Clean ↓ | Test Other ↓ | RTF ↓ |
|---|---|---|---|---|
| Transformer decoder | 38.5M | 6.39 | 17.03 | 1.52 |
| BLOCKDECODER | 37.9M | 6.19 | 16.94 | $0.66_{(\sim2.3x)}$ |
| Vanilla CTC-only | 37.9M | 10.05 | 23.87 | $0.36_{(\sim4.2x)}$ |
| InterCTC | 37.9M | 8.80 | 21.29 | $0.37_{(\sim4.1x)}$ |

## D Aggregate Attention Plots

Figure 6 presents aggregated self-attention and cross-attention plots computed over 250 utterances using the decoder from a model trained on the full Librispeech dataset. The observed patterns are consistent with those in Figure 1. To generate these plots, we first select utterances with a minimum audio length of $T_{\min}$ and a minimum label sequence length of $W_{\min}$ and then compute the average attention scores over the first $T_{\min}$ audio frames and $W_{\min}$ label positions.

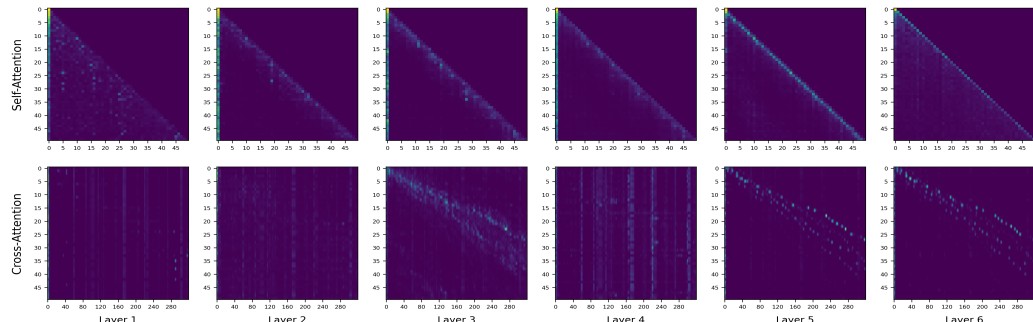

Figure 6: Attention plots illustrating the patterns learned by self-attention and cross-attention blocks across all layers of a standard Transformer decoder in a hybrid CTC/attention ASR system. The plot consists of two rows, one for self-attention and another for cross-attention and six columns, corresponding to the six decoder layers (ordered left to right). Brighter colors indicate higher attention weights. These visualizations are generated by aggregating attention scores over 250 utterances using the decoder of a model trained on the Librispeech 960h dataset.

## E Detailed Results on LibriSpeech 960h

Table 8 presents the complete results on Librispeech 960h dataset, with inference conducted on both CPU and GPU. A condensed version of these results is provided in Table 2. As outlined in Section 4.2, with CPU based inference, BLOCKDECODER achieves a nearly 2.5x speedup over the strongest baselines. However, in the case of GPU-based inference, the RTF values for BLOCKDECODER and the baseline are nearly identical. Through detailed profiling, we found that the majority of GPU inference time is spent on bookkeeping operations performed by the toolkit to support beam search, while the actual decoder forward passes contribute only a small fraction to the total inference time. Since

modifying the entire inference pipeline of the toolkit is both cumbersome and beyond the scope of this work, we additionally report the average total FLOPs (Floating Point Operations) and the mean decoder forward time incurred solely by the decoder during beam search. This offers a more focused assessment of the decoder's computational efficiency. As shown in Table 8, our proposed BLOCKDECODER reduces the FLOPs and the decoder forward time by approximately 60% compared to the baseline across both CPU and GPU-based inference.

Table 8: Comparison of the performance (WER %) of our system against other architectures on the full Librispeech dataset. Here, DFT stands for "Decoder Forward Time".

| Method | Params (M) | Common Metrics | | | CPU Inference | | GPU Inference | |
|---|---|---|---|---|---|---|---|---|
| | | Test Clean ↓ | Test Other ↓ | Average FLOPs (in TFLOPs) ↓ | RTF ↓ | Average DFT (in sec) ↓ | RTF ↓ | Average DFT (in sec) ↓ |
| **Transducer** | | | | | | | | |
| Transformer [21] | 139 | 2.4 | 5.6 | - | - | - | - | - |
| ContextNet [14] | 112.7 | 2.1 | 4.6 | - | - | - | - | - |
| Conformer (M) [4] | 30.7 | 2.3 | 5.0 | - | - | - | - | - |
| Conformer (L) [4] | 118.8 | 2.1 | 4.3 | - | - | - | - | - |
| **CTC** | | | | | | | | |
| QuartzNet (L) [35] | 19 | 3.9 | 11.3 | - | - | - | - | - |
| **Hybrid CTC+Attention** | | | | | | | | |
| Transformer [36] | 270 | 2.9 | 7.0 | - | - | - | - | - |
| Conformer [37] | 116.2 | 2.9 | 7.0 | - | - | - | - | - |
| Conformer (L) [6] | 147.8 | 2.2 | 4.7 | - | - | - | - | - |
| Branchformer [5] | 116.2 | 2.4 | 5.5 | - | - | - | - | - |
| Branchformer (L) [6] | 146.7 | 2.2 | 4.8 | - | - | - | - | - |
| E-Branchformer [6] | 148.9 | 2.1 | 4.5 | - | - | - | - | - |
| **Our baselines (Hybrid)** | | | | | | | | |
| E-Branchformer | | | | | | | | |
| — w/ Transformer Decoder | 148.9 | **2.1** | 4.6 | 3.08 | 7.73 | 53.24 | 0.338 | 0.232 |
| **Our work (Hybrid)** | | | | | | | | |
| — w/ BLOCKDECODER | 146.8 | **2.1** | **4.4** | **1.13**$_{(\sim2.7x)}$ | **2.98**$_{(\sim2.6x)}$ | **16.02**$_{(\sim3.3x)}$ | **0.306**$_{(\sim1.1x)}$ | **0.151**$_{(\sim1.5x)}$ |

# F   Implementation Details

The hyper-parameters used in all our experiments are reported in Table 9.

Table 9: A detailed summary of the configurations used for every dataset mentioned in our experiments.

| Hyper-parameters | Librispeech | | Tedlium2 | Aishell | Commonvoice | SLURP |
|---|---|---|---|---|---|---|
| | 100h | 960h | | | | |
| Frontend | | | | | | |
|   window length | 400 | 512 | 400 | 512 | 400 | 512 |
|   hop length | 160 | 160 | 160 | 128 | 160 | 128 |
| SpecAug | | | | | | |
|   time warp window | 5 | 5 | 5 | 5 | 5 | 5 |
|   num of freq masks | 2 | 2 | 2 | 2 | 2 | 2 |
|   freq mask width | $(0, 27)$ | $(0, 27)$ | $(0, 27)$ | $(0, 27)$ | $(0, 30)$ | $(0, 30)$ |
|   num of time masks | 5 | 10 | 5 | 10 | 2 | 2 |
|   time mask width | $(0, 0.05T)$ | $(0, 0.05T)$ | $(0, 0.05T)$ | $(0, 0.05T)$ | $(0, 40)$ | $(0, 40)$ |
| Architecture | | | | | | |
|   feature size ($d$) | 256 | 512 | 256 | 256 | 256 | 512 |
|   attention heads ($h$) | 4 | 8 | 4 | 4 | 4 | 8 |
|   num of encoder layers ($N_E$) | 12 | 17 | 12 | 12 | 12 | 12 |
|   encoder hidden size ($d_{\text{hidden}}^{\text{enc}}$) | 1024 | 1024 | 1024 | 1024 | 2048 | 1024 |
|   depth-wise conv kernel | 31 | 31 | 31 | 31 | 31 | 31 |
|   num of decoder layers ($N$) | 6 | 6 | 6 | 6 | 6 | 6 |
|   decoder hidden size ($d_{\text{hidden}}^{\text{dec}}$) | 2048 | 2048 | 2048 | 2048 | 2048 | 2048 |
| Training | | | | | | |
|   epochs | 70 | 80 | 50 | 60 | 50 | 60 |
|   learning rate | 2e-3 | 2e-3 | 2e-3 | 1e-3 | 1.0 | 1e-3 |
|   warmup steps | 15k | 40k | 15k | 35k | 25k | 35k |
|   weight decay | 1e-6 | 1e-6 | 1e-6 | 1e-6 | 1e-6 | 1e-6 |
|   Gradient Accum steps | 4 | 4 | 2 | 1 | 1 | 1 |
|   dropout rate | 0.1 | 0.1 | 0.1 | 0.1 | 0.1 | 0.1 |
|   ctc weight ($\delta$) | 0.3 | 0.3 | 0.3 | 0.3 | 0.3 | 0.3 |
|   label smoothing weight | 0.1 | 0.1 | 0.1 | 0.1 | 0.1 | 0.1 |

