# OpenReview forum: "BlockDecoder: Boosting ASR Decoders with Context and Merger Modules"
_NeurIPS.cc/2025/Conference — NeurIPS 2025 poster_

### Official Review · Reviewer_pHX8 · 2025-06-17

**Clarity:** 3
**Significance:** 4
**Originality:** 4
**Rating:** 5
**Confidence:** 4

**Summary:**

This paper introduces an approach to disentangle the self and cross attention layer in a transformer decoder for AED based ASR. This is based on an interesting observation by the authors that the initial layers of cross attention and the later layers of self-attention in a AED based ASR models are quite redundant. Based on this observation, the authors propose context and merger modules that disentangle the two aforementioned attention mechanisms. Furthermore, the merger module is equipped with the ability to perform block decoding which contributes to fast decoding capabilities of the proposed model as evidenced by the results.

**Questions:**

Here are some clarification questions I had while reading the paper:
1. In the equations between lines 105-106, there are residual connections after $LayerNorm_{att}$ and $LayerNorm_{ff}$ whereas no such residual connections are shown in figure 2(b). This can be a source of confusion for the reader.
2. Figure 2(a) was very confusing for me. It is not clear how to interpret it. Concretely, in Figure 2(a), it seems like $y_i ... y_{i+K-1}$ is conditioned on all previous tokens, but the equation in 3.2.2 suggest otherwise (the dependency seems to be only on $y_{i-1}$).
3. It is also unclear what the role/effect is of the CTC module in figure 2(a).
4. Figure 2(a) might benefit from a more detailed/thorough caption.
5. The authors' use of the modified token sequence suggests an increase in the training data to cover for all subsequences. Does this result in an increased training time? If so, how much? Furthermore, if there is an increase in training time, it raises the question of whether the proposed technique might be an overkill just to attain speed improvements and not much WER improvements?
6. The results suggest that the only improvements gained from the proposed model is in RTF/speedup. The authors should compare their model with state-of-the-art CTC models like self-conditioned,  hierarchically conditioned, BERT-based CTC models and self-conditioned folded encoders[1-4]. What is the speed up compared to these models and what is the corresponding trade-off with WER performance.
7. In appendix E, the authors show that their model is better than just disentangling self and cross attention in a standard transformer decoder. However, it is not clear why that is the case? Does the improvement come from the block decoding strategy or is there something else also?
8. I would like the authors to comment on whether it is possible to use a pre-trained language model or even an LLM as a substitute for the text encoder. Have the authors explored that direction?


[1] Nozaki et al. Relaxing the conditional independence assumption of CTC-based ASR by conditioning on intermediate predictions.

[2] Higuchi et al. Hierarchical conditional end-to-end ASR with CTC and multi-granular subword units.

[3] Higuchi et al. BERT meets CTC: New formulation of end-to-end speech recognition with pre-trained masked language model.

[4] Komatsu et al. Non-autoregressive asr with self-conditioned folded encoders.

**Ethical Concerns:**

["NO or VERY MINOR ethics concerns only"]

**Final Justification:**

This paper is built on a good research insight and furthers the advancements of inference speed in autoregressive models. Hence I suggest acceptance.

**Limitations:**

The authors have addressed limitations and my concerns to a large extent hence I would suggest acceptance of the paper which was my original recommendation at a score of 5.

**Paper Formatting Concerns:**

No concerns.

**Quality:**

3

**Strengths And Weaknesses:**

Strengths:
1. This paper builds on a good observation made on the working of transformer decoders which reflects good research insights.
2. The research contribution is significant in that the paper proposes a technique to further faster inference in ASR models.
3. The research contribution is original to the best of my knowledge.
4. Diverse set of experiments and ablation studies are conducted.

Weaknesses:
1. The paper has some clarity issues in properly explaining the technical details of their contribution (please see "Questions" section)
2. There is no significant WER improvements using the proposed model. Although the speed of inference improves, it is hard for me to see whether similar performance gains cannot be achieved by carefully training existing non-autoregressive models like CTC, self-conditioned CTC or hierarchically conditioned CTC models which are fully non-autoregressive and can do a very fast decoding.

---

> ### Author Rebuttal · Authors · 2025-07-31
>
> Dear Reviewer,
>
> Thank you for your insightful and constructive feedback! Below, we address each of your comments in detail:
>
> > In the equations between lines 105-106, there are residual connections after $LayerNorm_{att}$ and $LayerNorm_{ff}$ whereas no such residual connections are shown in figure 2(b). This can be a source of confusion for the reader.
>
> We deliberately omitted the residual connections in the figure to avoid visual clutter. This omission is mentioned in the Figure’s caption. We will revise the caption to clarify this further.
>
> > Figure 2(a) was very confusing for me. It is not clear how to interpret it. Concretely, in Figure 2(a), it seems like $y_i, \ldots y_{i+K-1}$ is conditioned on all previous tokens, but the equation in 3.2.2 suggest otherwise (the dependency seems to be only on $y_{i-1}$). Figure 2(a) might benefit from a more detailed/thorough caption.
>
> Thank you for this comment, and we apologize for the confusion. At each stage, the Merger receives representation of a progressively longer text prefix (e.g., [\<sos\>], [\<sos\>, _B] etc.) from Text-Encoder (shown via the blue bars) along with the corresponding audio representation, and predicts a block of three tokens auto-regressively. For example, in the first step, Merger is trained to predict three tokens conditioned on [\<sos\>], and in the second step, it is trained to predict the next three tokens conditioned on [\<sos\>, \_B], and so on. Since each block prediction is itself auto-regressive, we need a starting token. While works like Megabyte [1] use an underscore (\_)  as the first token in the block, we use the last token of the prefix (e.g., _B in the second step), which helps maintain continuity.
>
> We will revise both Figure 2 and the caption to make these points much clearer.
>
> [1] Yu, Lili, et al. "Megabyte: Predicting million-byte sequences with multiscale transformers." Advances in Neural Information Processing Systems 36 (2023): 78808-78823.
>
> > It is also unclear what the role/effect is of the CTC module in figure 2(a).
>
> We follow the hybrid CTC/Attention Encoder-Decoder framework [1], which uses a multi-task training objective. The CTC loss on the encoder encourages faster convergence and, during inference, helps guide the decoder to adhere to its monotonic alignments.
>
> [1] S. Watanabe, T. Hori, S. Kim, J. R. Hershey and T. Hayashi, "Hybrid CTC/Attention Architecture for End-to-End Speech Recognition," in IEEE Journal of Selected Topics in Signal Processing, vol. 11, no. 8, pp. 1240-1253, Dec. 2017, doi: 10.1109/JSTSP.2017.2763455.
>
> > The authors' use of the modified token sequence suggests an increase in the training data to cover for all subsequences. Does this result in an increased training time? If so, how much? Furthermore, if there is an increase in training time, it raises the question of whether the proposed technique might be an overkill just to attain speed improvements and not much WER improvements?
>
> Thank you for the suggestion. Sections 3.4 and 4.4 briefly discuss the impact of our architectural modifications on both training and inference time. To summarize: our experiments span datasets ranging from 100 to 1,000 hours of audio. Across this range, training time remains comparable to the baseline, while inference time is consistently reduced. We will revise the draft to better highlight these findings and add a more explicit summary of training/inference efficiency.
>
> > The results suggest that the only improvements gained from the proposed model is in RTF/speedup. The authors should compare their model with state-of-the-art CTC models like self-conditioned, hierarchically conditioned, BERT-based CTC models and self-conditioned folded encoders[1-4]. What is the speed-up compared to these models and what is the corresponding trade-off with WER performance.
>
> Thank you for this suggestion! While CTC-only models are generally faster due to greedy decoding, they often underperform compared to encoder-decoder models in terms of accuracy. Since our work adopts the encoder-decoder paradigm, we primarily compare against strong sequence-to-sequence baselines. Incorporating comparisons with advanced CTC-based methods is a promising direction, and we will aim to  include such comparisons in future work.
>
> > In appendix E, the authors show that their model is better than just disentangling self and cross attention in a standard transformer decoder. However, it is not clear why that is the case? Does the improvement come from the block decoding strategy or is there something else also?
>
> Yes, the improvement stems from the block-level design of the Merger. In the model from Appendix E, a single self-attention layer attempts to model both local (within block) and global (outside block) contexts simultaneously. Our design explicitly separates these requirements: self-attention in the block captures local structure, while cross-attention accesses broader text context. This modularity allows the Merger to independently learn and integrate these representations, improving expressivity and performance.
>
> > I would like the authors to comment on whether it is possible to use a pre-trained language model or even an LLM as a substitute for the text encoder. Have the authors explored that direction?
>
> Indeed, the modular structure of BlockDecoder naturally supports replacing the Text Encoder with a pre-trained language model, including LLMs. This is an exciting direction we intend to explore. Moreover, our architectural separation offers a pathway to apply pruning or distillation techniques to large models like Whisper to reduce inference cost, especially in streaming or real-time scenarios.
>
> Thanks again for your feedback, and please let us know if any further clarifications would be helpful.

---

> > ### Comment · Reviewer_pHX8 · 2025-08-05
> >
> > I thank the authors for their response to my review.
> >
> > I just have an additional comment regarding model comparisons. The reason I mentioned comparisons with CTC-based models is because one needs a compelling reason to use the proposed model. If the idea is to reduce the decoding time, one would obviously adopt versions of CTC-based models (which have achieved very good WER in the recent past owing to significant strides made in these models). Conversely, if the idea is to achieve very good WER, one would opt for autoregressive models. Since this paper tries to find a way to trade-off both, it stands to reason to compare with both these family of models to ascertain the degree to which we have emulated the advantages of these two families.

---

> > > ### Author Response · Authors · 2025-08-07
> > > **Additional CTC only experiment**
> > >
> > > Dear Reviewer,
> > >
> > > This is a great point. Your comment motivated us to evaluate a popular CTC-only variant, InterCTC [1], which is among the most widely used alternatives to full sequence-to-sequence decoders. We present the comparison below on Librispeech 100 hour:
> > >
> > > | Experiment | # of params | Test Clean (WER) | Test Other (WER) |
> > > | ----------- | ----------- | ----------- | ----------- |
> > > | E-branchformer w/ Transformer Decoder | 38.5M | 6.39 | 17.03 |
> > > | E-branchformer w/ intermediate CTC | 37.9M | 8.8 | 21.3 |
> > > | E-branchformer w/ Block Decoder | 38.1M | 6.19 | 16.94 |
> > >
> > > As seen above, the InterCTC-based model yields significantly worse performance compared to both our proposed BlockDecoder and the full Transformer decoder baseline. While InterCTC offers faster inference due to its CTC-only nature, it does so at a notable cost in recognition accuracy. This supports our claim that the BlockDecoder achieves a more favorable trade-off between latency and performance.
> > >
> > > Thank you for prompting this valuable experiment.
> > >
> > > [1]: Lee, Jaesong, and Shinji Watanabe. "Intermediate loss regularization for ctc-based speech recognition." ICASSP 2021-2021 IEEE International Conference on Acoustics, Speech and Signal Processing (ICASSP). IEEE, 2021.

---

### Official Review · Reviewer_w6fN · 2025-06-23

**Clarity:** 3
**Significance:** 3
**Originality:** 3
**Rating:** 4
**Confidence:** 5

**Summary:**

This paper introduces a novel decoder architecture for attention-based encoder-decoder ASR systems. The core idea is to disentangle the decoder into two specialized components: a text encoder block and an audio attention block, based on empirical observations. These are subsequently merged using a dedicated merge function. The architecture is designed in a block-wise manner, offering greater efficiency during inference compared to conventional decoder designs that rely on integrated self-attention and cross-attention mechanisms. Experiments on several standard ASR and SLU benchmarks demonstrate state-of-the-art performance, along with improved inference efficiency. Ablation studies further support the authors’ hypothesis that separating the text encoding and audio attention functions leads to both performance and efficiency gains.

**Questions:**

- It would be valuable to discuss how the proposed findings and architectural modifications could be extended to other models, such as Whisper—which uses a decoder with prompt conditioning—or decoder-only models. Such a discussion would enhance the generalizability and impact of the proposed method across different architectures.
- Similarly, is there relevant literature or comparable architectural separation in neural machine translation (NMT)? Discussions with NMT could broaden the relevance of the work.
- Please include a discussion of training time and scalability. Given the growing focus on model and data scalability in recent AI research, it would be helpful to understand how the proposed approach scales with respect to training time, data size, and model complexity.
- Could the text encoder block be replaced or initialized with a pre-trained large language model (LLM)? Leveraging LLMs may enhance the performance and attractiveness of the proposed architecture.
- In Section 3.1, line 82, the phrase “(CTC) loss [31] non-autoregressively” is misleading. While CTC inference is non-autoregressive, the CTC loss itself is computed using the forward-backward algorithm, and thus is not non-autoregressive in nature. Please clarify this distinction.
- In Section 3.2.1 (footnote), the expression $T=T'/S$ should specify that the result is rounded, since time frame division generally results in non-integer values.
- The equations describing the merge function are difficult to follow. They are currently unnumbered, and the associated explanations are hard to trace. It would help to number these equations and provide a clearer, step-by-step interpretation.
- The SLU task setup needs further clarification. For example, typical intent classification is a non-sequential task and does not require a decoder, whereas slot filling is sequential and does. If your SLU task combines ASR with intent classification, it would become a sequence-to-label problem. Please clarify the SLU formulation in the paper.
- In Section 4.4, the attention visualizations are informative. However, could you also include numerical metrics to quantify attention patterns? For instance, a diagonality score has been used in some studies to evaluate attention sharpness and alignment quality.

**Ethical Concerns:**

["NO or VERY MINOR ethics concerns only"]

**Final Justification:**

The authors’ responses address some of my concerns regarding clarifications, which will certainly improve the paper. However, they do not fully resolve the weaknesses I previously identified. In addition, these changes would require major revisions, making it difficult to properly assess the final paper at this stage. Therefore, I would like to maintain my score as Borderline Accept.

**Limitations:**

The authors acknowledge training time trade-offs, which appear to stem primarily from technical limitations. However, the paper does not address potential negative societal impacts. For instance, large-scale encoder-decoder ASR systems such as Whisper are known to exhibit hallucination issues due to the complexity and size of their decoder networks. Since the proposed method also relies on a decoder architecture, similar risks may be inherited. That said, the block-wise processing approach introduced here may help mitigate some of these issues. A discussion of such societal implications, especially in terms of reliability and misuse, would strengthen the broader impact assessment of the work.

**Paper Formatting Concerns:**

No concerns

**Quality:**

4

**Strengths And Weaknesses:**

Strengths
- The paper presents new insights into decoder behavior by highlighting the functional separation between text encoding and audio attention components.
- A novel decoder architecture is proposed based on this finding, utilizing a modular design with clear functional roles.
- The approach achieves efficient inference and demonstrates strong performance across multiple standard ASR and SLU benchmarks, supported by comprehensive ablation studies.

Weaknesses
- The motivation for adopting block-wise processing is not clearly justified. While it contributes to inference efficiency, it is unclear whether block processing is essential to the proposed architecture, as the core idea could potentially be applied to a standard decoder.
- The merge function is difficult to understand. The equation presented on page 4 is abstract, and the accompanying explanations are not fully grounded in the mathematical formulation.
- The experimental section lacks some practical implementation details, such as training time, hardware specifications (e.g., CPU type used for real-time factor measurements), and overall computational cost.
- The paper could benefit from a broader discussion of related work on decoder architectures. For instance, recent work exploring structured state space models and non-autoregressive decoding (e.g., mask prediction) are relevant and should be considered:
  - Miyazaki, Koichi, Masato Murata, and Tomoki Koriyama. "Structured state space decoder for speech recognition and synthesis." ICASSP 2023.
  - Higuchi, Y., Watanabe, S., Chen, N., Ogawa, T., & Kobayashi, T. "Mask CTC: Non-Autoregressive End-to-End ASR with CTC and Mask Predict." Interspeech 2020.

---

> ### Author Rebuttal · Authors · 2025-07-31
>
> Dear Reviewer,
>
> Thank you for your constructive and detailed feedback! Below, we address each of your comments in detail:
>
> > on extending the proposed architecture to Whisper, Decoder-Only Models, NMT, and Integration with Pretrained LLMs
>
> Thanks for these valuable suggestions! We agree that broadening the discussion to other areas/models/applications could further strengthen the impact of our work. While our proposed architecture is particularly effective in settings where the encoder output is substantially longer than the target sequence (e.g., ASR), similar dynamics occur in tasks such as document summarization. These often involve long input sequences and shorter outputs, making them promising candidates for adaptations of BlockDecoder. Additionally, since BlockDecoder separates the decoder into a text-only encoder and merger module, our framework naturally supports replacement of the text encoder with a pre-trained LLM, a direction we plan to explore in future work. Finally, the architectural modularity we introduce could inspire pruning strategies in large models such as Whisper to accelerate inference, especially in real-time use cases.
> We will add a discussion addressing these points in the revised draft.
>
> > Please include a discussion of training time and scalability. Given the growing focus on model and data scalability in recent AI research, it would be helpful to understand how the proposed approach scales with respect to training time, data size, and model complexity.
>
> Thank you for the suggestion. Sections 3.4 and 4.4 briefly discuss the impact of our architectural modifications on both training and inference time. To summarize: our experiments span datasets ranging from 100 to 1,000 hours of audio. Across this range, training time remains comparable to the baseline, while inference time is consistently reduced. We will revise the draft to better highlight these findings and add a more explicit summary of training/inference efficiency.
>
> > In Section 3.1, line 82, the phrase “(CTC) loss [31] non-autoregressively” is misleading. While CTC inference is non-autoregressive, the CTC loss itself is computed using the forward-backward algorithm, and thus is not non-autoregressive in nature. Please clarify this distinction.
>
> We respectfully clarify that CTC is indeed non-autoregressive. The CTC loss is computed using the forward-backward algorithm (that involves dynamic programming over time) to marginalize over all possible alignments but it does not introduce autoregressive dependencies (i.e., conditioning the output at time t on previous output tokens).
>
> > In Section 3.2.1 (footnote), the expression $T = T’ / S$ should specify that the result is rounded, since time frame division generally results in non-integer values.
> > The equations describing the merge function are difficult to follow. They are currently unnumbered, and the associated explanations are hard to trace. It would help to number these equations and provide a clearer, step-by-step interpretation.
>
> Thank you for these edits. We will revise the draft to incorporate these changes.
>
> > The SLU task setup needs further clarification. For example, typical intent classification is a non-sequential task and does not require a decoder, whereas slot filling is sequential and does. If your SLU task combines ASR with intent classification would become a sequence-to-label problem. Please clarify the SLU formulation in the paper.
>
> Apologies for the ambiguity. In our SLU setup, the model is indeed first trained with an ASR objective, where the output label sequence is augmented to include intent and entity related tags. At inference time, we decode the sequence as in ASR, and compute SLU metrics by parsing the decoded output. We will revise the SLU section to make this setup explicit.
>
> > In Section 4.4, the attention visualizations are informative. However, could you also include numerical metrics to quantify attention patterns? For instance, a diagonality score has been used in some studies to evaluate attention sharpness and alignment quality.
>
> Thank you for this suggestion! We will include diagonality scores in a revised version for a more quantitative measure of the attention patterns.
>
> Thanks again for your thoughtful review, and please let us know if any further clarifications would be helpful.

---

> > ### Comment · Area_Chair_KxBL · 2025-08-04
> >
> > Dear reviewer w6fN, please indicate whether the rebuttal clarified any of your questions.

---

> > ### Comment · Reviewer_w6fN · 2025-08-04
> >
> > Thanks for your answers.
> > These answers solve some of my concerns about the clarifications, which will definitely improve the paper.
> > However, they do not fully mitigate the weakness points that I raised, and I want to keep my score as it is.

---

> > > ### Author Response · Authors · 2025-08-07
> > > **Additional clarifications**
> > >
> > > Dear Reviewer,
> > >
> > > We realized that our initial rebuttal focused primarily on addressing the clarification questions you raised, and did not sufficiently respond to the weaknesses you identified. We would like to take this opportunity to address those points more directly:
> > >
> > > > The motivation for adopting block-wise processing is not clearly justified. While it contributes to inference efficiency, it is unclear whether block processing is essential to the proposed architecture, as the core idea could potentially be applied to a standard decoder.
> > >
> > > We named the architecture BlockDecoder as we wanted to highlight its ability to predict a block of tokens (auto-regressively). However, we would like to emphasize that our overall gains come from both predicting a block of tokens and decoupling the decoder into text-only (with self-attention) and text+audio (with cross-attention) modules. To further clarify: if the merger predicted only one token at a time, then the architecture, despite being modular, would incur similar computational costs to a standard decoder with early cross-attention layers removed (please refer to Appendix E for more details). Conversely, if a full decoder predicted tokens in blocks but retained all cross-attention layers, the speed-up would be limited, since the dominant cost arises from attending to long encoder outputs at every layer. Our approach combines modularity (removing early cross-attention) with block-level prediction, where only every k-th token attends to the text-encoder, reducing computation further by a factor of k. These two components are complementary: block prediction alone or modularity alone would yield partial gains, but together they enable the full efficiency benefits we report.
> > >
> > > > * The merge function is difficult to understand. The equation presented on page 4 is abstract, and the accompanying explanations are not fully grounded in the mathematical formulation.
> > > > * The experimental section lacks some practical implementation details, such as training time, hardware specifications (e.g., CPU type used for real-time factor measurements), and overall computational cost.
> > > > * The paper could benefit from a broader discussion of related work on decoder architectures. For instance, recent work exploring structured state space models and non-autoregressive decoding
> > >
> > > Apologies for the ambiguity. We will incorporate these changes in the final draft.
> > >
> > > Thank you again for your thoughtful review, and please let us know if any further clarifications would be helpful.

---

### Official Review · Reviewer_1eQS · 2025-06-26

**Clarity:** 3
**Significance:** 2
**Originality:** 3
**Rating:** 5
**Confidence:** 4

**Summary:**

A novel decoder architecture, BlockDecoder, is proposed for
attention-based encoder-decoder ASR. It is motivated after observing
that attention layers in conventional Transformed-based decoders first
focus on building textual context, and then on merging acoustic and
textual information. Based on this observation, BlockDecoder is
proposed as an alternative to conventional decoders. It consists of
two separate sub-modules, a text encoder with only self-attention
layers to build textual context, and a lightweight merger with
cross-attention layers fusing acoustic information from the encoder
and contextualized outputs from the text encoder. Empirical results
show that the proposed decoder roughly doubles in speed conventional
decoders without degradation in performance.

**Questions:**

* Could you please provide more evidence supporting the observation inspiring
  BlockDecoder? Please consider showing attention weights averaged over a set
  of examples instead of a single example.
* The conventional Transformer Decoder does not include efficient inference
  techniques. Could you please run additional experiments using these
  techniques? It would be good to see RTF results from these experiments in
  all relevant Tables.

**Ethical Concerns:**

["NO or VERY MINOR ethics concerns only"]

**Final Justification:**

I have updated my rating to 5 (accept) since my main doubts have been cleared up.

**Limitations:**

yes

**Paper Formatting Concerns:**

I have not noticed any major formatting issues in the paper.

**Quality:**

3

**Strengths And Weaknesses:**

**Strengths**
* Good work; its technical side and sound empirical assessment are of
  good quality. Improving the efficiency of cutting-edge ASR systems
  without performance degradation is challenging; and the authors
  manage to do so.
* The paper is largely clear.
* To my knowledge, the idea behind the proposed decoder is
  original. Speeding up transformer architectures is a hot research
  topic, and thus this work fits well within current research trends.

**Weaknesses**
* The observation inspiring BlockDecoder is based on limited evidence.
* The paper does not consider related work on efficient transformer inference
  such as KV Cache and FlashAttention. This renders significance to the main
  contribution.

---

> ### Author Rebuttal · Authors · 2025-07-31
>
> Dear Reviewer,
>
> Thank you for your valuable feedback. Below, we address your comments in detail:
>
> > Could you please provide more evidence supporting the observation inspiring BlockDecoder? Please consider showing attention weights averaged over a set of examples instead of a single example.
>
> Thanks for pointing this out!  We computed the average attention weights for all the attention blocks of the decoder across 1000 examples, and observed the same patterns shown in Figure 1. We will include this figure in the revised draft.
>
> > The conventional Transformer Decoder does not include efficient inference techniques. Could you please run additional experiments using these techniques? It would be good to see RTF results from these experiments in all relevant Tables.
>
> Thank you for raising this important point. We would like to clarify that all the results reported in our paper already incorporate widely used efficient inference techniques, including key-value (KV) caching and Automatic Mixed Precision (AMP). We will make sure to explicitly state this in the revised draft.
>
> Thanks again for your feedback, and please let us know if any further clarifications would be helpful.

---

> > ### Comment · Reviewer_1eQS · 2025-08-01
> > **FlashAttention**
> >
> > Have you used FlashAttention?

---

> ### Author Response · Authors · 2025-08-05
> **Addressing FlashAtention**
>
> Dear Reviewer,
>
> Apologies for the delayed response.
>
> > Have you used FlashAttention?
>
> FlashAttention is primarily designed to accelerate training by efficiently computing attention over batches of sequences on the GPU. However, during inference, particularly with techniques like KV-caching and beam search, it offers limited benefits. As our work focuses on fast inference, and the majority of our experiments are conducted using CPU-based inference (with the exception of full LibriSpeech evaluation), we initially did not incorporate FlashAttention.
> That said, for completeness, we re-implemented GPU-based inference for LibriSpeech using FlashAttention and we observe that inference-time performance with flash attention is comparable to the standard scaled dot-product attention, and the relative trends between the baseline and our proposed architecture remain consistent.
>
> Thank you again for your valuable feedback. Please let us know if any further clarification would be helpful.

---

### Official Review · Reviewer_zAsc · 2025-07-02

**Clarity:** 2
**Significance:** 2
**Originality:** 3
**Rating:** 5
**Confidence:** 4

**Summary:**

This is a very well-written, detailed proposal of a new architecture for end-to-end ASR that (1) separates text-only layers and audio+text layers in the decoder, and (2) predicts blocks of K tokens at a time, for faster inference. The architecture is motivated by insights from observing actual self- and cross- attention patterns in existing decoders, and by the desire for low-latency inference.

**Questions:**

Some comments/questions, intended as clarifying or constructive feedback.

L33: "This phenomenon has been previously observed in the AUDIOENCODER [5] but never analyzed in the DECODER".
Though not quite the same context, similar analysis of layer-wise transformer attention scores was done e.g. in

@inproceedings{Irie_2019, series={interspeech_2019},
   title={Language Modeling with Deep Transformers},
   url={http://dx.doi.org/10.21437/Interspeech.2019-2225},
   DOI={10.21437/interspeech.2019-2225},
   booktitle={Interspeech 2019},
   publisher={ISCA},
   author={Irie, Kazuki and Zeyer, Albert and Schlüter, Ralf and Ney, Hermann},
   year={2019},
   month=sep, collection={interspeech_2019} }

Fig. 1: what is the y-axis of each sub-plot? Tokens, I assume?

L37, "Based on these systematic attention patterns": given that so far this is analyzing the situation for just a single sample, it seems like a stretch to say these are "systematic" patterns.

L43, proposal of "BLOCK DECODER": I am wondering at this point if the "block" is conceptually the same as, or distinct from, the fact that a _block_ of tokens will be predicted. I.e., I could have a similar unit, call it a block, combining the two modules mentioned, that did not predict a block of tokens. I could also have a standard decoder, that does not distinguish text only from text+audio, that did predict a block of tokens. So upon first reading, I'm wondering whether the "block" in the name "block decoder" is actually helping the prediction of a "block" of tokens. Upon second reading, I see that the block decoder is indeed helping the prediction of a block of tokens -- could the authors make the presentation more easily accessible?

L47, "lesser" --> "fewer"

L59, "or introducing new training objectives [20, 21]". Great that the work is also considers neural transducers here and in the evaluation section. Transducers are more than a new training objective, though -- in fact, the transducer decoder, being purely text based, is quite strongly related to the proposed architecture in this submission. In a way a neural transducer is like the proposed architecture, but with a very simple Merger (the transducer "joiner"). This link could be discussed in more detail in a way that highlighted the merit of the proposed architecture.

Fig 2(b): these two blocks just seem like vanilla transformer stacks, one with self-attention, the other with cross-attention? If so, do we need to illustrate them? I note that the fonts in the figure, e.g. for "Audio Encoder" and "Text Encoder", are not consistent. Furthermore, the example, "_B L OCK" etc., propagates from Merger to merge by only apparently adding one new token, so this doesn't illustrate the key feature of the proposal, the prediction of multiple tokens?

L111 "dropouts" --> "dropout"

"Section 3.2.2 Merger": finally, getting to the equation (which should be numbered) at L124, and the text at L133, I get a sense for the details of the multi-token prediction. Since this is a key part of the proposal, ideally the basic approach here would be described more specifically, and much earlier in the paper, and as I suggested earlier, it should be made clearer how the multi-token prediction benefits from the BlockDecoder architecture.

**Ethical Concerns:**

["NO or VERY MINOR ethics concerns only"]

**Final Justification:**

In their rebuttal, the authors' showed good understanding of my central concerns regarding the key concepts and the manner of presentation, and indicated willingness to address those concerns -- particularly relating to expanded discussion of the connection with neural transducers, better presentation of the core and combined contributions of modular + block structure, improved notation and improved figures.

**Limitations:**

Yes.

**Paper Formatting Concerns:**

No concerns.

**Quality:**

3

**Strengths And Weaknesses:**

The core idea of leveraging separate text and text+audio blocks specifically to improve multi-token inference latency is interesting and apparently effective. A real strength of the work is its good writing and precise mathematical exposition, with further analysis of computational cost in the Appendix. Furthermore, a comprehensive set of WER evaluation is presented on some well-known standard test sets, with decent results -- matching, sometimes bettering, existing baselines, but more notably, producing improvements in inference latency.

While well-written and precise, the work is nonetheless often unclear in the bigger picture. Precise details can obscure the larger points, making the work slightly inaccessible. The first several pages feel like a rather long-winded exposition of the key aspects of the architecture. The reader really has to go over the proposal in detail to grasp how the proposed architecture is different from existing architectures. At first glance, it does not seem that original, e.g. reminds me of this study:

@misc{deng2023adaptableendtoendasrmodels,
      title={Adaptable End-to-End ASR Models using Replaceable Internal LMs and Residual Softmax},
      author={Keqi Deng and Philip C. Woodland},
      year={2023},
      eprint={2302.08579},
      archivePrefix={arXiv},
      primaryClass={eess.AS},
      url={https://arxiv.org/abs/2302.08579},
}

However, what I finally realized is that the proposal leverages this decoupling specifically for multi-token inference. That is a very interesting property -- but one that is not explained very clearly, and in fact not even mentioned in the Abstract.

It seems that overall the work could, ideally, be presented more effectively and with greater impact by significantly compressing the description of the aspects of the architecture that closely follow existing standard architectures, and highlighting the novel aspects, e.g. the multi-token prediction mechanism.

---

> ### Author Rebuttal · Authors · 2025-07-31
>
> Dear Reviewer,
>
> Thank you for your very thoughtful and constructive assessment of our work! Below, we address each of your comments in detail:
>
> > Fig. 1: what is the y-axis of each sub-plot? Tokens, I assume?
>
> Yes, the $y$-axis represents the token (or label) sequence.
>
> > L37, "Based on these systematic attention patterns": given that so far this is analyzing the situation for just a single sample, it seems like a stretch to say these are "systematic" patterns.
>
> Thanks for pointing this out!  We computed the average attention weights for all the attention blocks of the decoder across 1000 examples, and observed the same patterns shown in Figure 1. We will include this figure in the revised draft to be more precise when claiming systematic patterns.
>
> > L43, Clarifying the Role of "Block" in the Block Decoder and Its Comparison to Related Variants
>
> Thank you for this insightful comment!  The name BlockDecoder is indeed primarily meant to reflect its ability to predict a block of tokens (auto-regressively). However, we would like to emphasize that our overall gains come from both predicting a block of tokens and decoupling the decoder into text-only (with self-attention) and text+audio (with cross-attention) modules. To further clarify: if the merger predicted only one token at a time, then the architecture, despite being modular, would incur similar computational costs to a standard decoder with early cross-attention layers removed (please refer to Appendix E for more details). Conversely, if a full decoder predicted tokens in blocks but retained all cross-attention layers, the speed-up would be limited, since the dominant cost arises from attending to long encoder outputs at every layer. Our approach combines modularity (removing early cross-attention) with block-level prediction, where only every $k^{\text{th}}$ token attends to the text-encoder, reducing computation further by a factor of $k$. These two components are complementary: block prediction alone or modularity alone would yield partial gains, but together they enable the full efficiency benefits we report. We will revise the draft to clarify this dual role of “block” in the naming and motivation behind BlockDecoder.
>
> > L59,  On the Architectural Similarities and Differences with Neural Transducers
>
> We agree that our proposed architecture shares conceptual similarities with neural transducers, as both decouple the encoder and decoder streams and rely on a composition mechanism to integrate them. As the reviewer noted, the Joiner in the Transducer is a shallow feedforward network. In contrast, our Merger includes multiple attention layers, enabling richer and more flexible context integration. We will revise the draft to acknowledge this connection more clearly and highlight the architectural contributions of our approach.
>
> > Fig 2(b): these two blocks just seem like vanilla transformer stacks, one with self-attention, the other with cross-attention? If so, do we need to illustrate them? I note that the fonts in the figure, e.g. for "Audio Encoder" and "Text Encoder", are not consistent. Furthermore, the example, "_B L OCK" etc., propagates from Merger to merge by only apparently adding one new token, so this doesn't illustrate the key feature of the proposal, the prediction of multiple tokens?
>
> 1. The Merger is indeed designed to autoregressively predict a block of multiple tokens. At each stage, the Merger receives representation of a progressively longer text prefix (e.g., [\<sos\>], [\<sos\>, _B] etc.) from Text-Encoder (shown via the blue bars) along with the corresponding audio representation, and predicts a block of three tokens auto-regressively. We will revise both Figure 2 and the caption to make it clearer that multiple tokens are predicted.
> 2. We included the stacks in Fig 2b just to highlight that the Text Encoder contains only self-attention layers, while the Merger contains both self-attention and cross-attention layers. We will aim to visually highlight this better.
> 3. The font differences were intended to distinguish newly introduced components from pre-existing ones, but we will make the fonts consistent to avoid any confusion.
>
> > L33, prior work on attention (Irie et al.); Minor edits (L47 “lesser” → “fewer”, L111 “dropouts” → “dropout”; and section 3.2.2, the suggestion to present the core multi-token idea earlier and more clearly.
>
> Thank you for these detailed comments! We will revise the draft to incorporate these changes.
>
> Thanks again for your thoughtful review, and please let us know if any further clarifications would be helpful.

---

> > ### Comment · Area_Chair_KxBL · 2025-08-04
> >
> > Dear reviewer zAsc, please indicate whether the rebuttal clarified any of your questions.

---

> > > ### Comment · Reviewer_zAsc · 2025-08-05
> > >
> > > The rebuttal did clarify my questions, as I indicated my Final Justification.

---

### Note · Authors · 2025-08-12

We are very thankful to the reviewers for their thoughtful feedback and constructive discussions! We are very pleased that our work has received fairly positive ratings overall (one accept from reviewer pHX8 and three borderline accepts from reviewers zAsc, 1eQS, and w6fN). We will make sure to incorporate all the suggested edits and additional experiments into a revised version. Our main planned changes (apart from editorial fixes) are summarised below:
* We will place greater emphasis in the methodology section on the block decoding aspect of our proposed architecture and clarify how it improves upon decoder variants that use either a block-only structure or only a decoupled merger and text encoder.
* We will revise Figure 2 and its caption for improved clarity.
* We will include additional supporting evidence alongside Figure 1, such as averaged attention weights and a quantitative diagonality assessment, to further strengthen the claims underlying the BlockDecoder design.
* We will include experiments with CTC-only variants to illustrate that, while CTC-only models can achieve faster inference, they tend to sacrifice recognition accuracy; our preliminary experiments showed a notable degradation of up to 4% absolute WER.
* We will update the experimental setup section to provide more details on the spoken language understanding (SLU) task, the default inference techniques we used, and the hardware used for our experiments.
* We will add a paragraph in the analysis section discussing the broad applicability of our approach, including the use of LLMs as a substitute for the text encoder.

We once again thank the reviewers and area chairs for investing their valuable time and effort into reviewing our paper.

---

### Decision · Program_Chairs · 2025-09-17

**Decision:**

Accept (poster)

**Comment:**

This paper proposes a new architecture for ASR which uses text-only layers and audio+text layers and then predicts multiple outputs concurrently to speed up decoding. The reviewers praised the central idea, the good experimnetal results, and the efficient inference. Some of the reviewers concerns around limited discussion of other related work was addressed during the discussion period as well as the better presentation of the central ideas. Not all questions could be resolved during the discussion period, but I believe this is a good paper and I recommend acceptance.